# Influence of quantified dry cupping on soft tissue compliance in athletes with myofascial pain syndrome

Yen-Chun Chiu[1,2], Ioannis Manousakas[3], Shyh Ming Kuo[3], Jen-Wen Shiao[4], Chien-Liang Chen[5]*

1 Department of Orthopedics, E-Da Hospital/I-Shou University, Kaohsiung, Taiwan, 2 Department of Electrical Engineering, I-Shou University, Kaohsiung, Taiwan, 3 Department of Biomedical Engineering, I-Shou University, Kaohsiung, Taiwan, 4 Center for General Education, I-Shou University, Kaohsiung, Taiwan, 5 Department of Physical Therapy, I-Shou University, Kaohsiung, Taiwan

* chencl@isu.edu.tw

**Data Availability Statement:** All relevant data are within the manuscript.

**Funding:** This work is funded by the E-Da Healthcare Group of the Republic of China under

## Abstract

### Purpose

This study aimed to develop a quantitative dry cupping system that can monitor negative pressure attenuation and soft tissue pull-up during cupping to quantify soft tissue compliance.

### Methods

Baseball players with myofascial pain syndrome were recruited to validate the benefits of cupping therapy. Nine of 40 baseball players on the same team were diagnosed with trapezius myofascial pain syndrome; another nine players from the same team were recruited as controls. All participants received cupping with a negative pressure of 400 mmHg for 15 minutes each time, twice a week, for 4 weeks. Subjective perception was investigated using upper extremity function questionnaires, and soft tissue compliance was quantified objectively by the system.

### Results

During the 15-minute cupping procedure, pressure attenuation in the normal group was significantly greater than that in the myofascial group ($p = 0.017$). The soft tissue compliance in the normal group was significantly higher than that in the myofascial group ($p = 0.050$). Moreover, a 4-week cupping intervention resulted in an obvious increase in soft tissue lift in the myofascial pain group ($p = 0.027$), although there was no statistical difference in the improvement of soft tissue compliance. Shoulder ($p = 0.023$) and upper extremity function ($p = 0.008$) were significantly improved in both groups, but there was no significant difference between the two groups.

### Conclusion

This quantitative cupping monitoring system could immediately assess tissue compliance and facilitate the improvement of soft tissues after cupping therapy. Hence, it can be used in

the contract EDCHM108001. The grant was awarded to YCC and CLC. The funders had no role in study design, data collection and analysis, decision to publish, or preparation of the manuscript.

**Competing interests:** The authors have declared that no competing interests exist.

athletes to improve their functional recovery and maintain soft tissues health during the off-season period.

## Introduction

Myofascial pain syndrome is a common disorder in general medical practice [1]. It is a chronic regional pain syndrome caused by trigger points in the taut bands of skeletal muscle. Diagnosis usually depends on a comprehensive physical exam and clinical presentation. This condition is believed to be caused by muscle overuse, trauma, or psychological stress [2]. Common treatment methods include medications such as analgesic drugs and muscle relaxants, local injection with various kinds of medication, and physical therapy [3].

Cupping therapy is a popular alternative therapy that is used worldwide. Although the mechanisms behind the effectiveness of cupping therapy have been proposed by many authors, there is still no single theory to explain its effects entirely [4]. Even though it is used widely in the management of myofascial pain and in sports medicine [5, 6], there are still debates about the effectiveness in the current literature [7, 8]. One of the most commonly criticized issues is that cupping therapy is performed using traditional methods, and the dose of cupping, included number of cups, negative pressure value, and duration and frequency of cupping cannot be quantified. Among these, the negative pressure value is the most difficult to define. Traditionally, it depends solely on the operator's experience or a rough estimation of soft tissue elevation in the cup [9]. Studies have shown that the marked cupping ecchymosis is caused by pressure ranging from -400 to -700 hPa (equivalent to -300 to -525 mmHg) [10, 11]. Our previous study compared the effects of autonomic nerve responses caused by cupping on the back with different negative pressures (-100, -300, and -500 mmHg) [12]. However, the above-mentioned pressures only represent data measured at the beginning of cupping. Since soft tissue compliance is not the same in all individuals and may be affected by various diseases, the difference between soft tissue pull-up and negative pressure attenuation during cupping (usually 10–15 minutes) needs to be quantified. This would ensure an "effective dose" of cupping. This unquantified procedure not only causes experimental errors, but also makes the study unreliable.

Nowadays, the cups used for cupping are mostly made of clear plastic material, usually polycarbonate. Different sizes of cups are used depending on the treatment site. Each cup has an air valve at the top where a manual pump can be connected and the air inside the cup can be pumped out. Using this type of equipment, there is no indicator of the pressure level achieved in the cup or any type of leakage. The pressure applied depends on the experience of the user. Thus, quantitative, objective, and repeatable studies cannot be performed.

The purpose of our study was to improve upon the drawbacks of traditional methods by developing a quantitative dry cupping equipment. We also applied this equipment to elite baseball players to compare soft tissue compliance during cupping in normal individuals and those with myofascial pain syndromes. The effectiveness of cupping therapy in relieving myofascial pain and improving functional outcomes was evaluated.

## Materials and methods

### Participants

Forty elite male baseball players from I-Shou University without comorbidities or contraindication for cupping therapy were enrolled. The mean age was 19.28±0.24 years, height was 176.0±1.5 cm, and weight 75.11±3.46 kg. Since the athletes' training intensity may affect the myofascial condition, we recruited all participants (including the experimental group and the

control group) from the same team to avoid training differences that may interfere with the effects of cupping intervention. Informed consent was obtained from all participants before the study started. This study was approved by the Institutional Review Board (EMRP-108-017) of E-Da Hospital, and performed in the spirit of the Helsinki Declaration.

In each of the 40 participants, a physician examined the trapezius muscles of the neck and upper back to identify any myofascial trigger points (MTrPs). As there is still no consensus in the diagnostic criteria of myofascial pain syndrome [13, 14], the diagnosis was made by MTrP identification through comprehensive physical examination. Doctors diagnosed nine baseball players with myofascial pain syndrome. Therefore, we set those nine participants as the experimental group and recruited nine healthy volunteers from the same team as the control group. There was no significant difference in height, weight and age between the two groups.

## Experimental design

The experiment was conducted during the off-season period to avoid worsening symptoms due to participation in intense competition. In the first week, all baseball players were examined by a physician and allocated into either the experimental group or control group. All participants completed the questionnaires about upper extremity function before cupping intervention (Fig 1).

In the second week, all participants started a 4-week cupping therapy program. During this 4-week period, the athletes were required to undergo dry cupping intervention with a negative pressure of 400 mmHg after routine training on each experimental day, for up to 15 minutes each time, twice a week, at least two days apart. Six cups were used for cupping therapy, and the cups were placed at the upper, middle, and lower fibers of the trapezius muscles on both sides (Fig 2). The position of the cups was defined based on the distribution map of common myofascial points in the upper back. Most of the nine participants in this study who suffered from myofascial pain syndrome had symptoms in the upper fiber, and only some of them had symptoms in the middle fiber or the lower fiber. Considering the consistent symptoms in the myofascial group, this study only evaluated the soft tissue of the upper trapezius muscle fibers on the affected side in the myofascial group and the dominant side in the normal group.

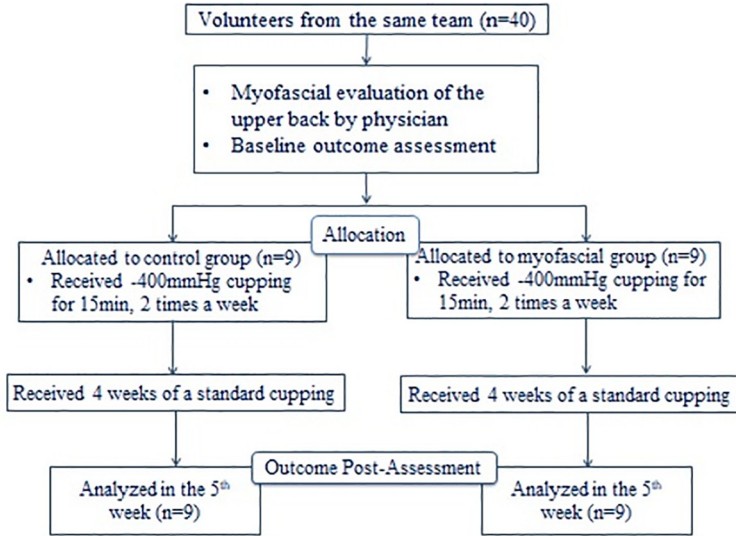

**Fig 1. Flowchart of the experimental procedure.**

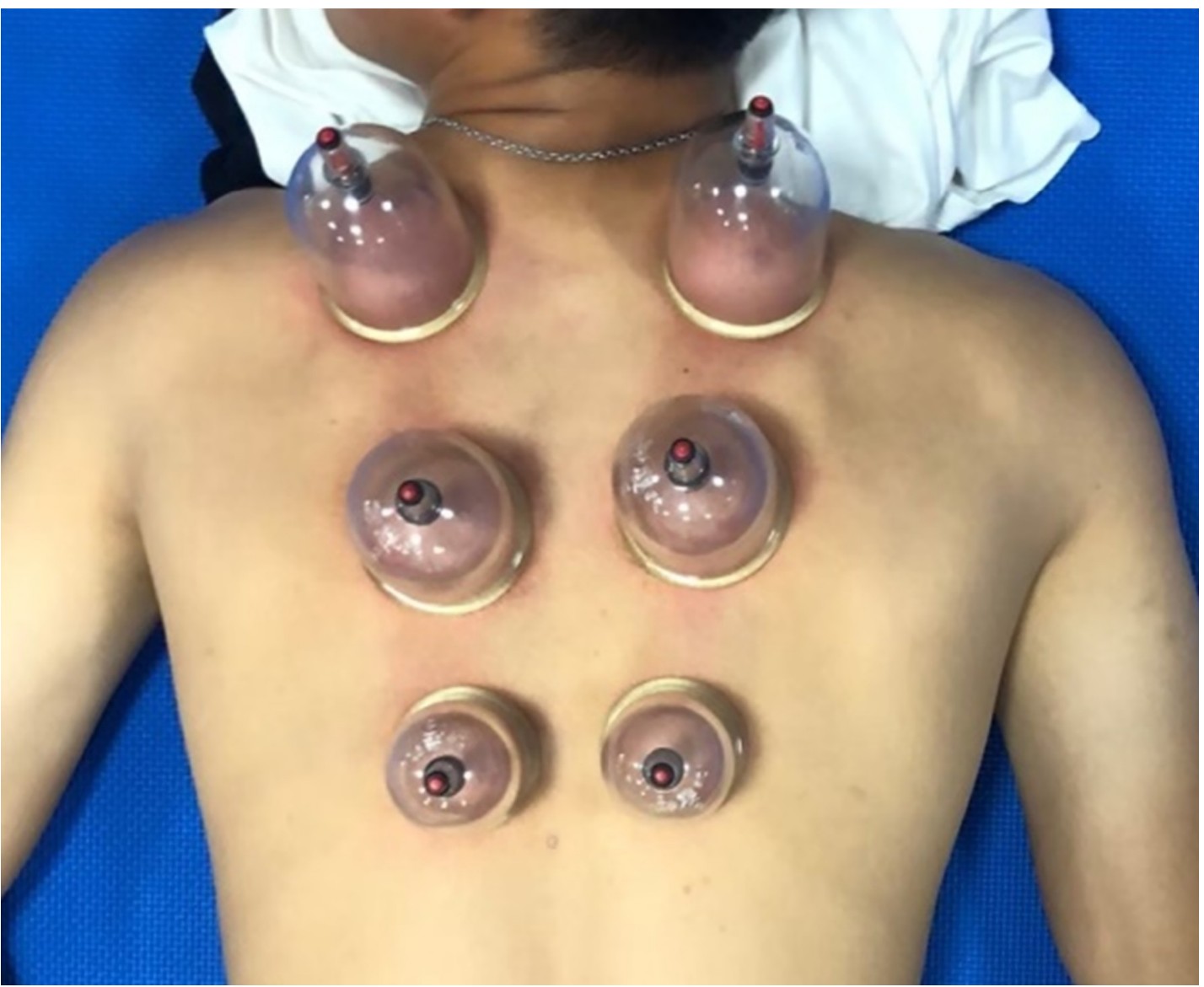

**Fig 2. Vacuum applications during dry cupping.** The individual in this manuscript has given written informed consent (as outlined in PLOS consent form) to publish these case details.

However, cupping therapy was applied to all fibers. The real-time negative pressure and height of soft tissue elevation were recorded. Soft tissue compliance (mm/mmHg) was defined as the height of soft tissue elevation divided by the negative pressure [15]. Moreover, the upper extremity function questionnaires were retested at the end of cupping to evaluate the effect of cupping therapy. One week after finishing the cupping therapy program, all participants were re-evaluated by the same physician to review the condition of myofascial pain syndrome. Post-test functional outcomes were assessed at the same time.

## Questionnaires for upper extremities and shoulder function

For people suffering from myofascial pain syndrome in the trapezius region, the function of the upper extremities or shoulder is indirectly affected. Therefore, this study also used

subjective questionnaires to assess the function of the upper extremities and shoulder complex in the normal and myofascial pain groups. The two questionnaires used are described below:

**Disability of Arm, Shoulder, and Hand (DASH).** The Chinese version of the DASH questionnaire is an upper limb assessment questionnaire that includes 30 questions. The score relates to functional activities that require the use of the upper extremities. The remaining questions include the following: two items related to pain, three related to other symptoms, one related to social life, one directly related to work, and one related to sleep. It also determines the patient's perception of his or her abilities. The score is based on a 100-point scale. A higher score indicates higher severity.

**Flexilevel Scale of Shoulder Function (FLEX-SF).** FLEX-SF was used to identify symptoms of dysfunction. In this scale, participants first answered a question which strictly classified their level of functionality as low, medium, or high. They then only responded to specific items targeted to their level of functionality. Scores ranged from 1 (representing the most limited function) to 50 (representing full functionality).

## Establishment of a monitor system for quantifying dry cupping

A system that can accommodate six cups was constructed. Six commercially available cups with a 6.5-cm outer diameter and a 5.5-cm inner diameter were acquired (Shen-Nong Cupping R0416179, Income Instrument Co., Ltd., Taiwan). For this study, the requirements were to undertake continuous measurement of the pressure in each cup as well as a measurement of the soft tissue elevation. The pressure range requirement was from 0 to -500 mmHg (-67 kPa). The pressure sensors that were selected (PSE533; SMC, Tokyo, Japan) have a pressure range from 100 kPa to -100 kPa. They require a power supply of 12 to 24 volts and provide an analog output proportional to pressure from 1 to 5 volts. The sensors are reported by the manufacturer to have an accuracy of less than ±2% of the full scale (FS), linearity of less than ±1% FS, and repeatability of less than ±1% FS. A pressure sensor is shown in Fig 3(A). The pressure sensors were tested for accuracy using a clinical mercury sphygmomanometer (CK-101; Spirit, New Taipei City, Taiwan). The available pressure range was from 0 to -280 mmHg. Measurements were repeated three times and mean and standard deviation values were calculated. The maximum absolute error of the mean values was 2.3 mmHg which is within the sensors specifications.

The majority of the distance sensors require contact with the skin. Contactless distance sensors were mostly optical. Although very accurate sensors exist, they are expensive and bulky. Recent technology has presented affordable optical sensors based on the Time-of-Flight (ToF) technology, which measures the time light takes to travel to an object in the angle of view of the sensor and reflect back to the sensor. Here, ToF sensors (VL6180X, STMicroelectronics, Geneva, Switzerland) were employed since they are small ($4.8 \times 2.8 \times 1.0$ mm) and inexpensive. They can measure distances at a range of 0 to 10 cm, independently of target reflectance. The sensor combines an infrared emitter and receiver. The control of the sensor, as well as distance reading, are performed using an $I^2C$ digital interface. The manufacturer states that the sensor measurement values have a noise of 2 mm maximum, which is defined as the maximum standard deviation of 100 measurements. The distance sensors were tested for accuracy using a carton disk mounted on a z-axis slide with a Vernier scale of 0.1mm per division (ZWG60, Misumi, Tokyo, Japan). The disk was elevated inside each cup from 0 mm (the rim of the cup) to a height of 37 mm within the cup with a maximum step of 5 mm. Each measurement was performed 30 times and for all the six cups. The average and standard deviation of the measurements were calculated. The maximum standard deviation for the measurements was 1.6 mm and the maximum absolute error of the mean values was 2 mm. A 3D drawing of a

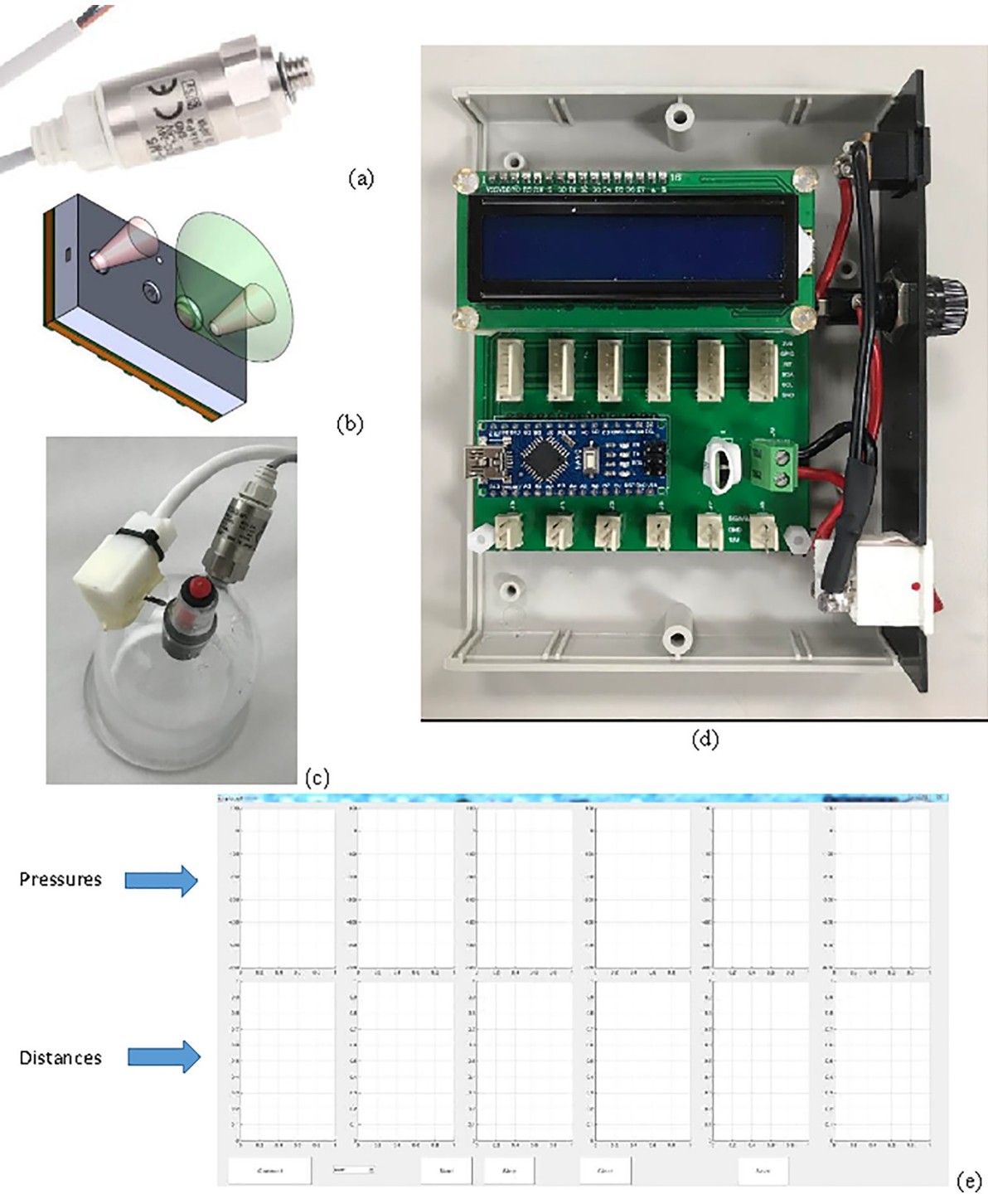

**Fig 3.** Photographs of hardware (a) a pressure sensor, (b) drawing of the field of view of the distance sensor, (c) suggested solution for the distance sensor cover, (d) photograph of the main device with the cover removed, and (e) a screen capture of the software graphical user interface.

distance sensor and its field of view are shown in Fig 3(B). To protect the sensors from dirt and condensed humidity, which can affect the performance of the sensor, as well as to protect

the sensor from low negative pressures, the sensors were protected within custom designed cases with a glass window and an opaque barrier between the field of view of the emitter and the receiver. This solution is recommended by the sensors' manufacturer. The cases were manufactured using resin 3D printing.

All the sensors were wired to a custom-made main device, which included a microprocessor board, an LCD screen, a DC to DC converter, and an external power supply. The microprocessor board (Arduino Nano or any other equivalent) was responsible for collecting data and redirecting the data to a notebook PC. The LCD screen displayed the current pressure measurements continuously for all six sensors. The DC to DC converter converted the voltage provided by the power supply to the voltages required by the sensors and the microprocessor board. A program written in MATLAB (MATLAB, Release 2014b, The MathWorks, Inc., Natick, MA, USA) was written to provide communication between the PC and the microcontroller, as well as to control and collect the sensors' data. The sensors' data were displayed in real time as pressure and distance graphs vs. time. The software could also save the data in comma-separated value files so that they could be easily imported into any other software for statistical analysis.

The pressure sensors were mounted on the cups using threaded holes. Openings on the cups were drilled for the distance sensors' infrared beams to have a direct view of the middle area of the cups without any distortions passing through the curved plastic material of the cups. A hand-held infrared viewer (ElectroViewer 7215; Electrophysics Corporation, Fairfield, NJ, USA) was used to view and adjust the ToF sensors' infrared beams so that they pointed to the center of the cups. The sensor cases were then fixed on the cups using epoxy resin. A 3D drawing of the sensor case, as well as a cup with the attached sensors, is shown in Fig 3(C). The main device was constructed as seen in Fig 3(D). It connected to a PC via a USB cable. A screen capture of the software graphical user interface is shown in Fig 3(E).

## Statistical analyses

The study was powered to detect an effect size of the primary outcome measure of 0.6, which was estimated based on the findings of a pilot study on dry cupping in myofascial pain syndrome [16]. To detect this effect with 80% power and a two-sided α of 0.05, a sample of eight patients was needed for each group. Data are presented as the mean and standard error of the mean (mean ± SEM). The generalized estimating equation (GEE) was used to analyze the time effect, and the trial-by-time and group-by-time interactions. The main analyses were subjective perceptions of upper extremity function (including DASH and FLEX-SF questionnaires) and objective soft tissue changes (including tissue elevation and tissue compliance). The improvement in DASH or FLEX-SF scores in the myofascial group and the normal group after the 4-week cupping intervention program was compared (group-by-time interaction). Moreover, changes in cup pressure and soft tissue elevation in the myofascial group and the normal group when undergoing the 15-min cupping intervention (group-by-time interaction) were compared. We also assessed whether the changes after 4 weeks (1st trial vs. 4th trial) were different between the two groups (trial-by-time interaction). Statistical analyses were conducted using PASW Statistics 23.0 (SPSS Inc., Chicago, IL, USA) software. The level of significance was set at $p < 0.05$.

## Results

After 4 weeks of cupping treatment, nine patients suffering from myofascial pain were evaluated by the same physician again. The MTrPs could not be identified in eight cases, and treatment was ineffective in one case. The total effective rate was 88.9%. Below, we describe the

**Table 1.  Effect of 15-minute cupping on trapezius soft tissue in the myofascial pain group and the normal group.**

| Parameters (mean ± SEM) | Normal (n = 9) | | Myofascial (n = 9) | | P value (Waldχ²) | |
|---|---|---|---|---|---|---|
| | Initial cupping | End of cupping | Initial cupping | End of cupping | Time | Group × time |
| **Pressure** (mmHg) | 402.40 ± 0.74 | 173.65 ± 16.41 | 401.99 ± 0.38 | 222.82 ± 13.06 | <0.001 (199.415) | 0.017 (5.658) |
| **Distance** (mm) | 17.94 ± 0.553 | 32.56 ± 4.307 | 20.39 ± 1.82 | 28.94 ± 1.68 | 0.001 (10.944) | 0.183 (1.775) |
| **Compliance** (mm/mmHg) | 0.044 ± 0.002 | 0.244 ± 0.056 | 0.050 ± 0.005 | 0.138 ± 0.011 | <0.001 (12.617) | 0.050 (3.851) |

The level of significance was set at $p < 0.05$.

difference between the soft tissue response of the myofascial pain group and the normal group during 15-minute cupping and the improvements after 4 weeks of cupping therapy.

## Immediate effect on trapezius soft tissue during 15-minute dry cupping and the differences between the two groups (myofascial vs. normal groups)

During the 15-minute cupping process, the overall main effect of the pressure values gradually decreased from the initial cupping to the end of cupping in both groups ($p < 0.001$). However, the degree of soft tissue lifting progressively increased during this 15-minute period ($p = 0.001$). The negative pressure application induced a fast component of tissue deformation at the initial cupping followed by a slow component, which lasted until the end of the cupping application. Assuming the tissue deformation was due to a translocation of fluid into the negative pressurized tissue, the tissue compliance was calculated. This study found that the overall main effect of the compliance values increased significantly ($p < 0.001$) from the initial cupping to the end of cupping in both groups (Table 1).

In terms of the attenuation range of the pressure in the cup in the two groups during 15-minute cupping (group-by-time interaction), it was found that the pressure attenuation in the normal group was significantly greater than that in the myofascial group ($p = 0.017$) (Table 1). However, the distance that the soft tissue was pulled up during the 15-minute cupping was not statistically different in the myofascial group and the normal group ($p = 0.183$). At the end of 15 minutes of cupping, a similar soft tissue pulling effect was achieved with a lower negative pressure in the normal group, so the soft tissue compliance in the normal group was significantly higher than that in the myofascial group ($p = 0.050$) (Fig 4).

## Effect on trapezius soft tissue after cupping for 4 weeks in the myofascial pain group

Analysis of the effect of cupping for 4 weeks on the pressure change in the cup for 15 minutes (trial-by-time interaction) showed that the slope of the pressure attenuation in the cup ($p = 0.038$) in the normal group was more obvious in the fourth week than in the first week. However, there was no significant difference between soft tissue lifting height ($p = 0.640$) and soft tissue compliance ($p = 0.209$) between the first week and the fourth week of cupping (Table 2). On the other hand, a 4-week cupping intervention resulted in an obvious increase in soft tissue lift in the myofascial pain group ($p = 0.027$) (Fig 5). Although there was no significant difference in the improvement of soft tissue compliance ($p = 0.296$) and the degree of pressure attenuation in the cups ($p = 0.228$) (Table 2), they all tended to approach the values of the normal group (Figs 4 and 5). In other words, we found that the increase in the compliance slope of the normal group during the 15-minute cupping period was significantly greater than that of the myofascial group (Fig 4C). However, we found that after 4 weeks of cupping

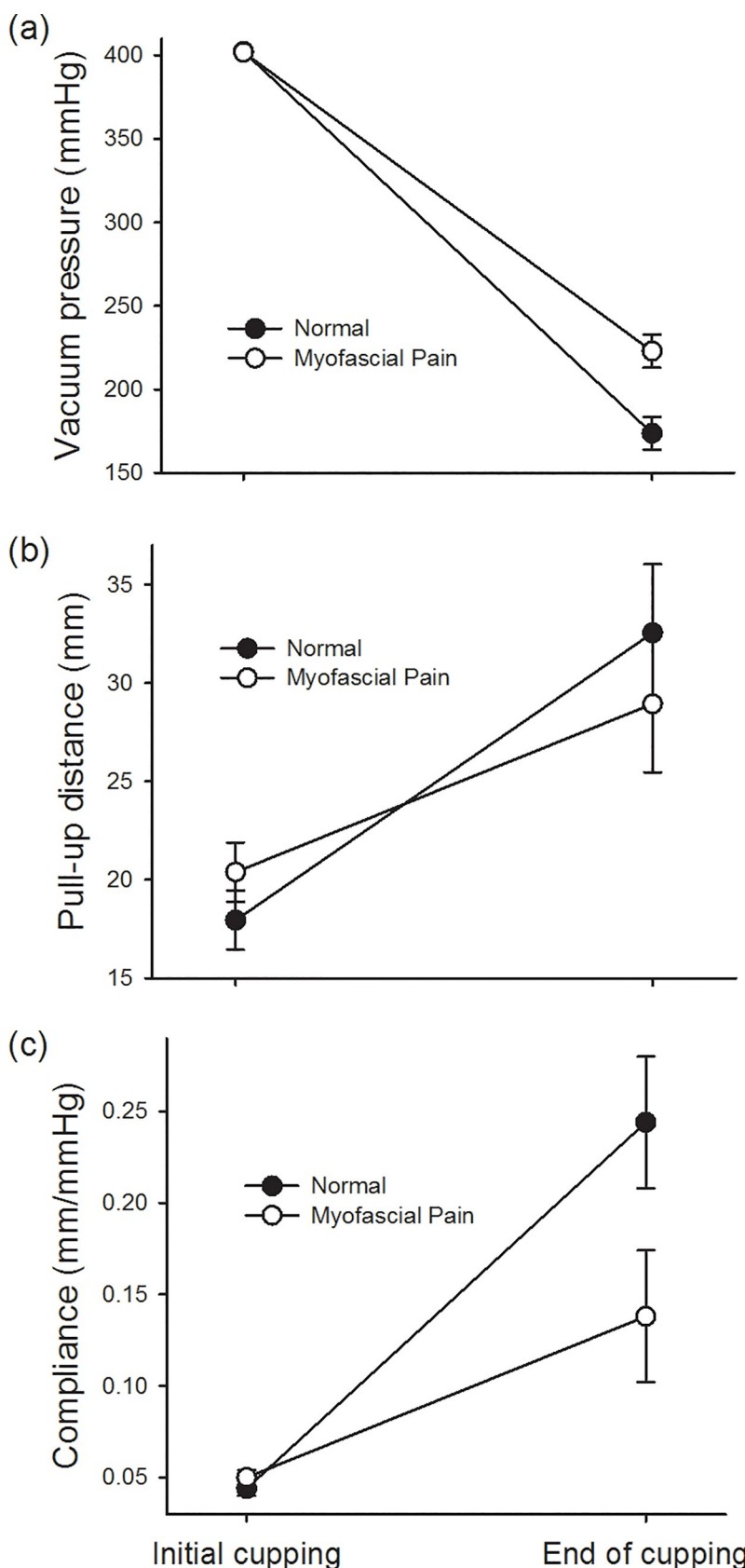

**Fig 4. Effect of 15-minute cupping on trapezius soft tissue in the myofascial pain group and normal group.** (a) The pressure attenuation in the cup and (c) the soft tissue compliance are statistically different in the two groups ($p = 0.017$ and $p = 0.050$); (b) however, there is no significant difference between the two groups in term of soft tissue lifting ($p = 0.183$).

treatment in the myofascial group, the slope of compliance had a tendency to increase as high as that in the normal group, although it was still not statistically significant (Fig 5C).

## Physical function of the upper extremity after 4-week cupping therapy in the myofascial pain group and normal group

Physical function of the upper extremities, as assessed by the DASH scale, showed a significant improvement ($p = 0.008$). Shoulder function (FLEX-SF) also improved significantly ($p = 0.023$) (Table 3). However, there was no statistical difference in the interaction between the myofascial group and the normal group after cupping for 4 weeks (group-by-time interaction). The results showed that both the experimental group and the control group subjectively perceived the positive effects of the 4-week cupping program. The improvement in subjective upper extremity function scores supported the trend toward normalization of soft tissue compliance.

## Discussion

The quantitative cupping system developed in this study can effectively monitor the pressure changes and soft tissue lifting response during cupping and compare the differences between myofascial pain and normal soft tissue. As there are no previous studies setting accuracy and reliability specifications and the standard deviation values of the accuracy measurements are smaller than the standard deviation values in the experimental group measurements, we could state that the accuracy and reliability are sufficient for this pilot study. During the 15-minute cupping process, the soft tissue was gradually lifted, accompanied by the attenuation of the pressure in the cup. This was represented by an increase in soft tissue compliance. However, the pressure attenuation in the cup in the myofascial pain group was significantly lesser than that in the normal group, and the increase in soft tissue compliance was significantly smaller in the myofascial pain group than in the normal group (Fig 4). Cupping therapy for 4 weeks increased the soft tissue pull-up height in the myofascial pain group. Although there was no statistical difference in improving the soft tissue compliance and pressure attenuation in the cup, these measurements tended to be more similar to those in the normal group (Fig 5). After 4 weeks of cupping therapy on the upper trapezius, participants reported significant

**Table 2. Effect of 4-week cupping therapy on trapezius soft tissue in the myofascial pain group and the normal group.**

| Parameters (mean ± SEM) | week (trial) | Normal (n = 9) | | P value (Waldχ²) | | Myofascial (n = 9) | | P value (Waldχ²) | |
|---|---|---|---|---|---|---|---|---|---|
| | | Initial cupping | End of cupping | Time | Trial × time | Initial cupping | End of cupping | Time | Trial × time |
| **Pressure** | 1st | 401.8±0.3 | 203.3±19.0 | <0.001 | 0.038 | 402.2±0.7 | 238.2±17.6 | <0.001 | 0.228 |
| (mmHg) | 4th | 403.0±1.4 | 144.0±22.8 | (110.24) | (4.32) | 401.8±0.3 | 207.4±17.8 | (84.52) | (1.46) |
| **Distance** | 1st | 18.44±0.71 | 31.00±4.82 | 0.008 | 0.640 | 17.67±1.19 | 28.33±1.32 | <0.001 | 0.027 |
| (mm) | 4th | 17.44±0.82 | 34.11±7.10 | (7.007) | (0.22) | 23.11±3.20 | 29.56±3.08 | (94.04) | (4.91) |
| **Compliance** | 1st | 0.044±0.002 | 0.177±0.042 | 0.001 | 0.209 | 0.043±0.003 | 0.123±0.008 | <0.001 | 0.296 |
| (mm/mmHg) | 4th | 0.043±0.002 | 0.311±0.098 | (10.12) | (1.58) | 0.057±0.008 | 0.153±0.021 | (81.00) | (1.09) |

The level of significance was set at $p<0.05$.

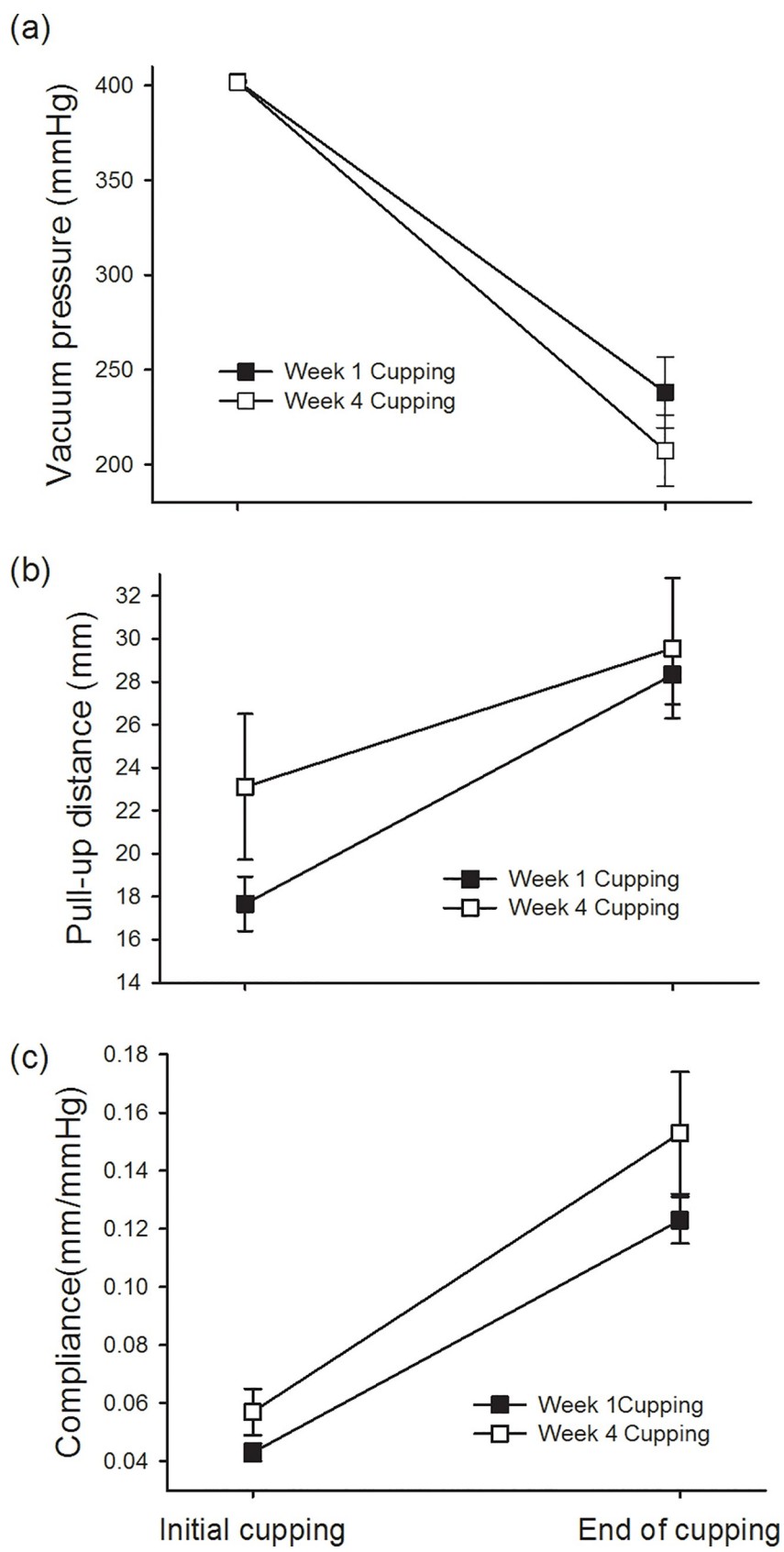

**Fig 5. Effect of 4-week cupping therapy on soft tissue of athletes with myofascial pain syndrome.** (b) The height of soft tissue pull-up during cupping at week 4 is significantly greater than that at week 1 ($p = 0.027$); however, there is no statistical difference between (a) pressure attenuation in the cup and (c) compliance of soft tissue at week 1 and 4.

improvements in upper extremity (DASH score) and shoulder function (FLEX-SF score) on the questionnaire, but there was no statistical difference between the two groups.

Compared with the FLEX-SF scores, the DASH scores show a more consistent trend in the effect of cupping on physical function and muscle compliance (Table 3). This may be due to the extensive assessment of DASH, which not only assesses upper extremity function, but also pain and sleep quality. Previous studies have confirmed that patients with myofascial pain syndrome experience a significant impact on pain and sleep quality [17, 18]. The effectiveness of cupping for improving pain and sleep quality has also been proven [19]. In contrast, the FLEX-SF score in the myofascial group barely increased after treatment for 4 weeks. The limitation of FLEX-SF may be because the FLEX-SF scores are mainly used to assess shoulder extension/flexion, internal/external rotation, abduction/adduction, or a combination of body axis or diagonal movement in daily life. These movements involve complex shoulder girdle integration activities. It is not easy to expect that these complex movements of the myofascial group can be significantly improved through simple trapezius cupping and short-term intervention. Nevertheless, the FLEX-SF score in the normal group increased significantly, and this result seemed to be explained by the results of previous studies. Kim et al. (2017) showed that cupping increased muscle flexibility and range of motion in healthy people, and its effect was similar to passive stretching [20]. In addition, the FLEX-SF scores in the normal and myofascial groups seemed to be the same at those at baseline in this study. It may be that the shoulder dysfunction of all participants was not serious, so the difference in scores between the two groups was not obvious at the beginning. When filling in FLEX-SF, the participant's shoulder function was divided into three levels: high, medium, and low. In this study, 16 participants (88.9%) were high-level function, and only two (11.1%) were medium level.

Several previous studies have confirmed that dry cupping can improve the symptoms of myofascial pain syndrome and restore activity [5, 21], and our findings were consistent with those of these studies. However, the studies mostly used subjective clinical symptoms to evaluate the effect of cupping on improving myofascial pain, and our study was characterized by a combination of subjective questionnaires and quantitative assessments of objective soft tissue compliance. This is the first study to use a self-developed soft tissue compliance evaluation system to immediately assess soft tissue compliance changes during cupping and the short-term changes in soft tissue after 4 weeks of cupping.

Myofascial pain syndrome is the existence of painful taut bands of muscle that contain discrete, hypersensitive foci. A clear mechanistic understanding of the disorder does not exist. This is likely due to the complex nature of the disorder, which involves the integration of

**Table 3. Comparison of upper extremity function between the myofascial pain group and the normal group after cupping for 4 weeks.**

| Upper extremity function | Group | Baseline | 4th week | P value (Wald$\chi^2$) | |
|---|---|---|---|---|---|
| | | | | Time | Group × time |
| **DASH** | Normal (n = 9) | 2.50±0.88 | 1.31±0.53 | 0.008 (6.971) | 0.222 (1.491) |
| | Myofascial (n = 9) | 6.11±2.24 | 2.87±1.49 | | |
| **FLEX-SF** | Normal (n = 9) | 35.89±0.90 | 38.67±0.31 | 0.023 (5.198) | 0.099 (2.726) |
| | Myofascial (n = 9) | 36.11±1.13 | 36.56±1.53 | | |

DASH = Disability of Arm, Shoulder and Hand; FLEX-SF = Flexilevel Scale of Shoulder Function; The level of significance is $p < 0.05$.

excitation-contraction coupling, neuromuscular inputs, local circulation, and energy metabolism. Cupping may be involved in regulating some of these mechanisms, which could improve the severity of myofascial pain syndrome. Previous studies have found that dry cupping may help reduce deoxy-hemoglobin and to obtain more oxy-hemoglobin, which enhances local oxygen uptake and promotes blood microcirculation and hemodynamic activity [22]. This effect may help improve myofascial pain syndrome and facilitate muscular function. Our previous research supported this inference, and we found that cupping during the recovery period of intense exercise could accelerate the recovery of muscle fatigue, and thus maintain better exercise performance [23]. The above evidence can be used to explain why participants in this study, whether normal or suffering from myofascial pain syndrome, reported a significant improvement in physical function after cupping.

The hyperactivity of the sympathetic nervous system plays an important role in the pathophysiology of myofascial pain syndrome [24, 25]. Some studies have indicated that self-massage combined with home exercise can improve parasympathetic activity [26], and dry needling treatment can effectively reduce the sympathetic skin response [27]. Both the above interventions can help improve myofascial pain syndrome. Our previous research also confirmed that cupping of the upper back can improve the effect of parasympathetic activity in heart rate variability [12]. Therefore, the treatment should have similar effects and mechanisms as dry needling and massage therapy. This may also explain why a subjective functional improvement was also apparent in the normal group in this study.

This study had several limitations. First, there is no universally accepted criteria for the diagnosis of myofascial pain syndrome [13, 14]. Hence, we defined the condition only through the existence of MTrP based on the physical examination in our study. Furthermore, we merely judged the treatment response by assessing the identification of MTrP. This may have influenced the reliability in terms of participant enrollment and evaluation of treatment outcomes without standard criteria. Second, in order to avoid interference with cupping therapy due to differences in training courses and intensity, this study chose baseball players from the same team as participants. However, because there were only nine players suffering from myofascial pain syndrome, the small sample size is considered one of the limitations of this study. Third, this study was performed in the off-season phase in order to reduce the probability of sports injury interference. In addition, the difference in the assignment of pitchers and fielders during the competition may also cause unequal exercise fatigue and interfere with experimental control variables. In the off-season phase, the duration of research intervention was quite limited. During the experiment, the athletes' routine training was controlled by the coaching team for 4 weeks. Longer-term interventions may render the effect of cupping more obvious.

The quantitative cupping monitoring system developed in this research has potential to be used as an index parameter in the future to verify whether the any improvement in soft tissue compliance. Our quantitative dry cupping equipment is safe, effective, and easy to operate. Hence, it can be used for elite athletes undergoing routine training to improve the recovery of shoulder function and to maintain healthy soft tissues during the off-season period.

## Conclusions

Our study applied a dry cupping system to elite baseball players with myofascial pain syndrome to distinguish the differences between their tissue compliance and that of a normal group. In addition, MTrP could not be identified in 89% of patients with myofascial pain after 4 weeks of cupping, and an improvement in the soft tissues were improved, which was confirmed by the results of the self-perception questionnaire. By quantitative cupping and observation of objective soft tissue changes, the pathological state of the soft tissue can be evaluated

in patients with myofascial pain and improvements in the soft tissues after cupping therapy can be observed. Use of scientific cupping and soft tissue evaluation can quantify the cupping dose and observe changes in tissue compliance, which is different from the traditional cupping method that is quantified by practitioner experience only.

## Acknowledgments

We thank all participating athletes and baseball team administrator Shih-Hung Hsu.

## Author Contributions

**Conceptualization:** Chien-Liang Chen.

**Data curation:** Ioannis Manousakas, Chien-Liang Chen.

**Formal analysis:** Chien-Liang Chen.

**Funding acquisition:** Yen-Chun Chiu, Chien-Liang Chen.

**Investigation:** Yen-Chun Chiu.

**Methodology:** Ioannis Manousakas, Chien-Liang Chen.

**Project administration:** Yen-Chun Chiu, Shyh Ming Kuo, Jen-Wen Shiao.

**Resources:** Shyh Ming Kuo, Jen-Wen Shiao.

**Software:** Ioannis Manousakas.

**Supervision:** Shyh Ming Kuo, Jen-Wen Shiao, Chien-Liang Chen.

**Validation:** Ioannis Manousakas.

**Writing – original draft:** Yen-Chun Chiu, Ioannis Manousakas, Chien-Liang Chen.

**Writing – review & editing:** Shyh Ming Kuo, Chien-Liang Chen.

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
