## [Decision Letter · Decision Letter 0]

7 Aug 2020

PONE-D-20-15383

Quantified dry cupping in baseball players with myofascial pain syndrome

PLOS ONE

Dear Dr. Chen,

Thank you for submitting your manuscript to PLOS ONE. After careful consideration, we feel that it has merit but does not fully meet PLOS ONE’s publication criteria as it currently stands. Therefore, we invite you to submit a revised version of the manuscript that addresses the points raised during the review process.,

In general, this is an interesting study, and the most important and novel contribution of this study is devising a system that can quantitatively monitor the “dose” and effect of cupping therapy. However, based on the comments provided by all reviewers, I will suggest the authors to make a major revision for a further evaluation to see whether the quality meet the publication criteria of the journal.,

We look forward to receiving your revised manuscript.

Kind regards,

Yi-Hung Liao, Ph.D.

Academic Editor

PLOS ONE

Journal Requirements:

2. Please refrain from stating p values as 0.00, either report the exact value or employ the format p<0.001.

3. We note that Figure [2] includes an image of a [patient / participant / in the study]. 

Additional Editor Comments (if provided):

In general, this is an interesting study, and the most important and novel contribution of this study is devising a system that can quantitatively monitor the “dose” and effect of cupping therapy. However, based on the comments provided by all reviewers, I will suggest the authors to make a major revision for a further evaluation to see whether the quality meet the publication criteria of the journal.

Reviewers' comments:

Reviewer's Responses to Questions

**Comments to the Author**

1. Is the manuscript technically sound, and do the data support the conclusions?

Reviewer #1: Yes

Reviewer #2: No

Reviewer #3: Partly

2. Has the statistical analysis been performed appropriately and rigorously? 

Reviewer #1: Yes

Reviewer #2: No

Reviewer #3: Yes

3. Have the authors made all data underlying the findings in their manuscript fully available?

Reviewer #1: Yes

Reviewer #2: Yes

Reviewer #3: Yes

4. Is the manuscript presented in an intelligible fashion and written in standard English?

Reviewer #1: Yes

Reviewer #2: Yes

Reviewer #3: Yes

5. Review Comments to the Author

Reviewer #1: Overall, the authors show very interesting results. They applied a new cupping machine on assuaging the myofascial syndrome. The introduction provided the relevant and sufficient information for the reader to under the idea of the work. Nevertheless, to polish the article, the authors may elaborate more on some paragraph.

Specific comments follow.

Material and method:

P.4 line 94-95: in this part, the authors claimed that 9 baseball players were diagnosed with myofascial syndrome. However, the definition or criteria of myofascial syndrome remained unclear. The authors may provide more info herewith.

Result:

P10 line 209-211: It remains ambiguous that how authors defined “8 cases were cured.” Please consider rephrasing by adding more detail.

Regarding table 3, the score of FLEX-SF of normal and myofascial group seemed to be the same at basal. In addition, the score of myofascial group barely increased after receiving treatment for 4 weeks. One paragraph regarding the limitation of FLEX-SF is recommended to be added (in discussion). On the other hand, the result of DASH showed consistency with muscle compliance.

Conclusions:

P16 line 339: Please provide more info regarding “8 cases were cured” in the result section, otherwise saying “some patients were cured” in the conclusion might be an overstatement.

P17 line 342-343: The authors seem to attempt to term muscle compliance as an indicator for the pathological state of soft tissue. To enhance that, a scatter plot or correlation analysis of muscle compliance and DASH might be an option.

Reviewer #2: Here are some questions:

1. Why conventional therapy was applied after intervention therapy? And the follow-up conventional therapy was not discussed in your study?

2. Why only upper fibers of trapezius were evaluated ?

3. How do your sample size been set up in your study design? The sample size seems insufficient to support the statistical hypothesis of this study. Under the statistical method with RM-ANOVA, the normal distribution is necessary. Should you consider using nonparametric statistics instead of RM-ANOVA analysis?

Reviewer #3: This study raises important questions and suggests a way to address them. Although cupping is widely used, the quantitative analysis of its efficacy has been limited. The authors introduced a system that can resolve this issue to some extent, which should be considered as an achievement worth publishing. Here, I suggest some modifications.

In my opinion, the most important and novel contribution of this study is devising a system that can quantitatively monitor the “dose” and effect of cupping therapy. (Although the authors also presented the results of the designed therapy, the outcomes were just consistent with those of previous studies, which reported the positive effects of cupping.) If the integration of such a system is the main contribution, the reliability and validity of the system should be clarified and emphasized. In lines 278-279, the authors actually claimed “The quantitative cupping system developed in this study can effectively monitor the pressure changes and soft tissue lifting response during cupping.” The authors show that both the pressure sensors and the optical sensors were working in some ways, but I could not find any solid evidence that the integrated system can provide sufficiently accurate and reliable measurement. Authors specified the reliability of the sensors in lines 146 – 162, but as they clarified themselves, the integrated system should work well under condensed humidity and non-negligible negative pressure. To avoid any malfunction, they also customized cases to protect the sensors. Then, it is naturally expected for the authors to verify whether the whole integrated system as well as the cases work well. The audience, including me, may expect that the measurement from the integrated system, which employs compact and inexpensive sensors, should be compared with any gold standard; although the accuracy and reliability of the system may not be perfect, are they good enough to quantify the “dose” and “effect” of the cupping therapy? If so, clearer justification (more than the specification from the manufacturer of each itemized hardware) is needed.

Regarding the outcome of the designed (and quantitatively monitored) therapy, further clarification can improve the manuscript. The authors claimed that the soft tissue compliance of the experimental group approached the values of the normal group after cupping for 4 weeks (lines 249-251). Does it mean that the difference in compliance values between the experimental group and the control group is not statistically significant after the 4 weeks? Thorough comparisons between the two groups after 4 weeks therapy will be helpful for the readers to understand the effect of the designed therapy.

In addition to these major suggestions, I have some minor comments:

1. I recommend that the authors should change the title so that it can reflect their main contribution. For example, the word “baseball player” does not seem important enough to be emphasized in the title.

2. Some sentences read awkward and seem to employ incorrect prepositions. (For example, the sentence starting from line 174 could be written much more concisely.) I recommend that the authors receive professional editing service.

3. Only after I read the manuscript for a while, I could understand that the control group is also from the same baseball team. This fact should be clarified much earlier.

4. The sentence in line 217-218 is very confusing. The current sentence reads that the tissue elevation reduced the pressure, which cannot be valid. The tissue elevation reduces the “magnitude of negative pressure”, or increases the absolute pressure value in the cup. The authors need to rephrase the sentence to clarify the meaning.

5. The sentence in lines 220-222 implies that compliance should increase or decrease due to the mechanical input to the tissue. This cannot be valid. Compliance (or the inverse of the stiffness) is the mechanical property of any system like human tissue. Mechanical input and output (e.g., force and kinematics or vice versa) can be measured to estimate this property. Although the compliance might be nonlinear and change due to the input, the specific tendency cannot be assumed before the actual estimation. If the authors defined compliance correctly, the phrase following “because” in the mentioned sentence cannot be the cause of the significant increase of the compliance.

6. The last statement starting from line 343 is clearly an overstatement. Either the suggested quantifying system or other more advanced system and the resulting quantified “dose” of cupping cannot “guarantee” the efficacy of cupping.

6. PLOS authors have the option to publish the peer review history of their article (what does this mean?). If published, this will include your full peer review and any attached files.

Reviewer #1: **Yes: **Sheng-Ju CHUANG

Reviewer #2: No

Reviewer #3: No

---

## [Author Response · Author response to Decision Letter 0]

13 Sep 2020

Thank you for your constructive and positive review of our manuscript entitled “Quantified dry cupping in baseball players with myofascial pain syndrome”. Our responses are included in the following pages. Below, we have addressed all of your comments on a point-by-point basis. We hope that the revisions are in order.

The revised manuscript has been checked by an expert in English writing and prepared according to the journal instructions, and we hope that you will now find it to be satisfactory for publication in your esteemed journal. We would like to thank you in advance for your consideration of the manuscript.

Responses to Journal Requirements

1. Thank you for updating your data availability statement. You note that your data are available within the Supporting Information files, but no such files have been included with your submission. At this time we ask that you please upload your minimal data set as a Supporting Information file, or to a public repository such as Figshare or Dryad.

Response: In the last submission, we mistakenly planted part of the message, now it is corrected to “All relevant data are within the manuscript and there are no other Supporting Information files”. 

2. We note that Figure [2] includes an image of a participant in the study.

Response: We have obtained the consent of the participant in the photo, and have added the description "The individual in this manuscript has given written informed consent (as outlined in PLOS consent form) to publish these case details" in the section of Materials and Methods. Lines 125-126 of page 6.

Responses to Reviewers’ Comments

Reviewer : 1

METHODS:

1. P.4 line 94-95: in this part, the authors claimed that 9 baseball players were diagnosed with myofascial syndrome. However, the definition or criteria of myofascial syndrome remained unclear. The authors may provide more info herewith.

Response: We greatly appreciate the reviewer’s comment. 

As we know, myofascial pain syndrome is defined as a regional pain disorder caused by the presence of trigger points within muscles or their fascia. Actually, there is no universally accepted diagnostic criteria till now (Ref [13, 14]). In our study, we made the diagnosis of myofascial pain syndrome depends on the presence of myofascial trigger points through a comprehensive physical exam by an orthopaedic doctor (P.5, lines 92-94 of the revised manuscript). It is a drawback in our manuscript hence we add a paragraph to describe it in the discussion section (P.18, lines 370-375 of the revised manuscript). Two appropriate references have been cited.

[13] Giamberardino MA, Affaitati G, Fabrizio A, Costantini R. Myofascial pain syndromes and their evaluation. Best Pract Res Clin Rheumatol. 2011;25(2):185-98.

[14] Yap E-C. Myofascial pain-an overview. Ann Acad Med Singapore. 2007;36(1):43.

RESULTS:

1. P10 line 209-211: It remains ambiguous that how authors defined “8 cases were cured.” Please consider rephrasing by adding more detail.

Response: We thank the reviewer for his/her comments. As your opinion, using the word “cured” seems inappropriate. We had modified our manuscript and replacing it with “MTrP could not be identified”. We believed that it will be more suitable and meet your recommendation. (P.10, line 226; P.18, lines 370-375). 

2. Regarding table 3, the score of FLEX-SF of normal and myofascial group seemed to be the same at basal. In addition, the score of myofascial group barely increased after receiving treatment for 4 weeks. One paragraph regarding the limitation of FLEX-SF is recommended to be added (in discussion). On the other hand, the result of DASH showed consistency with muscle compliance.

Response: We are deeply appreciated for these valuable suggestions, a paragraph about DASH and FLEX-SF in Table 3 has been added to the discussion section (P.16, lines 320-339). The text is as follows:

Compared with the FLEX-SF scores, the DASH scores show a more consistent trend in the effect of cupping on physical function and muscle compliance (Table 3). This may be due to the extensive assessment of DASH, which not only assesses upper extremity function, but also pain and sleep quality. Previous studies have confirmed that patients with myofascial pain syndrome experience a significant impact on pain and sleep quality (Ref [17, 18]). The effectiveness of cupping for improving pain and sleep quality has also been proven (Ref [19]). In contrast, the FLEX-SF score in the myofascial group barely increased after treatment for 4 weeks. The limitation of FLEX-SF may be because the FLEX-SF scores are mainly used to assess shoulder extension/flexion, internal/external rotation, abduction/adduction, or a combination of body axis or diagonal movement in daily life. These movements involve complex shoulder girdle integration activities. It is not easy to expect that these complex movements of the myofascial group can be significantly improved through simple trapezius cupping and short-term intervention. Nevertheless, the FLEX-SF score in the normal group increased significantly, and this result seemed to be explained by the results of previous studies. Kim et al. (2017) showed that cupping increased muscle flexibility and range of motion in healthy people, and its effect was similar to passive stretching (Ref [20]). In addition, the FLEX-SF scores in the normal and myofascial groups seemed to be the same at those at baseline in this study. It may be that the shoulder dysfunction of all participants was not serious, so the difference in scores between the two groups was not obvious at the beginning. When filling in FLEX-SF, the participant’s shoulder function was divided into three levels: high, medium, and low. In this study, 16 participants (88.9%) were high-level function, and only two (11.1%) were medium level. 

[17] Kim SA, Yang KI, Oh KY, Hwangbo Y. Association between sleep quality and myofascial pain syndrome in Korean adults: questionnaire based study. J Musculoskelet Pain. 2014;22(3):232-6.

[18] Muñoz-Muñoz S, Muñoz-García MT, Alburquerque-Sendín F, Arroyo-Morales M, Fernández-de-las-Peñas C. Myofascial trigger points, pain, disability, and sleep quality in individuals with mechanical neck pain. J Manipulative Physiol Ther. 2012;35(8):608-13.

[19] Volpato MP, Breda IC, de Carvalho RC, de Castro Moura C, Ferreira LL, Silva ML, et al. Single Cupping Thearpy Session Improves Pain, Sleep, and Disability in Patients with Nonspecific Chronic Low Back Pain. J Acupunct Meridian Stud. 2020;13(2):48-52. 

[20] Kim J-E, Cho J-E, Do K-S, Lim S-Y, Kim H-J, Yim J-E. Effect of cupping therapy on range of motion, pain threshold, and muscle activity of the hamstring muscle compared to passive stretching. Korean Society of Physical Med. 2017;12(3):23-32.

CONCLUSIONS

1. P16 line 339: Please provide more info regarding “8 cases were cured” in the result section, otherwise saying “some patients were cured” in the conclusion might be an overstatement.

Response: Thank you for your thoughtfulness bringing this to our attention. We have followed your suggestion to replace “cured” with “MTrP could not be identified” in the manuscript (P19, line 393) and stated its limitation in the discussion section (P18, lines 370-375).

2. P17 line 342-343: The authors seem to attempt to term muscle compliance as an indicator for the pathological state of soft tissue. To enhance that, a scatter plot or correlation analysis of muscle compliance and DASH might be an option.

Response: We greatly appreciate the reviewer’s comment. After analyzing the correlation between soft tissue compliance and DASH score, it was found that the correlation between the two did not reach statistical significance. In the present study, sample size of participants with myofascial pain syndrome may have been a limiting factor. Given this limitation, a larger sample size must be evaluated in future researches to confirm the validity of this indicator.

Reviewer: 2

1. Why conventional therapy was applied after intervention therapy? And the follow-up conventional therapy was not discussed in your study?

Response: We greatly appreciate the reviewer’s comment.

The field of this study was not in the medical institution, so the participants did not receive conventional therapy, only cupping interventional therapy. Therefore, there was no follow-up analysis of conventional therapy.

2. Why only upper fibers of trapezius were evaluated?

Response: We thank the reviewer for his/her comments.

The trapezius muscle has a larger area and is divided into three areas: upper, middle and lower fibers. Patients with myofascial pain syndrome mostly occur in the upper fibers, while the middle and lower fibers are relatively fewer. Most of the nine participants in this study who suffered from myofascial pain syndrome had symptoms in the upper fiber, and only some of them had symptoms in the middle fiber or the lower fiber. Considering the consistent symptoms in the myofascial group, this study only evaluated the soft tissue of the upper trapezius muscle fibers on the affected side in the myofascial group and the dominant side in the normal group. However, cupping therapy was applied to all fibers. In order to make the reader understand this reason more clearly, we have added some descriptions in lines 112-117 of the revised manuscript.

3. How do your sample size been set up in your study design? The sample size seems insufficient to support the statistical hypothesis of this study. Under the statistical method with RM-ANOVA, the normal distribution is necessary. Should you consider using nonparametric statistics instead of RM-ANOVA analysis?

Response: We understand all the suggestions that reviewer has made to make the article statistically rigorous, and we are grateful for it. 

1). “The study was powered to detect an effect size of the primary outcome measure of 0.6, which was estimated based on the findings of a pilot study on dry cupping in myofascial pain syndrome (Ref [16]). To detect this effect with 80% power and a two-sided α of 0.05, a sample of eight patients was needed for each group.” The preceding statement has now been added to the text (P10, lines 208-211 of the revised manuscript).

[16] Nasb M, Qun X, Ruckmal Withanage C, Lingfeng X, Hong C. Dry cupping, ischemic compression, or their combination for the treatment of trigger points: a pilot randomized trial. J Altern Complement Med. 2020;26(1):44-50.

2). We greatly appreciate the reviewer’s comment. Yes, we indeed agreed to change the statistical method of RM-ANOVA and use the generalized estimating equation (GEE) to analyze this research data with a smaller sample size. The modification caused by changing the statistical method is shown in line 212, Tables 1-3 and Figs 4-5, and their corresponding texts.

Reviewer: 3

Major suggestions:

1. In my opinion, the most important and novel contribution of this study is devising a system that can quantitatively monitor the “dose” and effect of cupping therapy. (Although the authors also presented the results of the designed therapy, the outcomes were just consistent with those of previous studies, which reported the positive effects of cupping.) If the integration of such a system is the main contribution, the reliability and validity of the system should be clarified and emphasized. In lines 278-279, the authors actually claimed “The quantitative cupping system developed in this study can effectively monitor the pressure changes and soft tissue lifting response during cupping.” The authors show that both the pressure sensors and the optical sensors were working in some ways, but I could not find any solid evidence that the integrated system can provide sufficiently accurate and reliable measurement. Authors specified the reliability of the sensors in lines 146 – 162, but as they clarified themselves, the integrated system should work well under condensed humidity and non-negligible negative pressure. To avoid any malfunction, they also customized cases to protect the sensors. Then, it is naturally expected for the authors to verify whether the whole integrated system as well as the cases work well. The audience, including me, may expect that the measurement from the integrated system, which employs compact and inexpensive sensors, should be compared with any gold standard; although the accuracy and reliability of the system may not be perfect, are they good enough to quantify the “dose” and “effect” of the cupping therapy? If so, clearer justification (more than the specification from the manufacturer of each itemized hardware) is needed.

Response: We are deeply appreciated for these valuable suggestions. 

As there are no previous studies setting accuracy and reliability specifications and the standard deviation values of the accuracy measurements are smaller than the standard deviation values in the experimental group measurements, we could state that the accuracy and reliability are sufficient for this pilot study. (P15, lines 305-308). 

In addition, we also conducted some tests on the accuracy of the pressure and distance sensors. They are described as follows:

1). P7-8, lines155-159: The pressure sensors were tested for accuracy using a clinical mercury sphygmomanometer (CK-101; Spirit, New Taipei City, Taiwan). The available pressure range was from 0 to -280 mmHg. Measurements were repeated three times and mean and standard deviation values were calculated. The maximum absolute error of the mean values was 2.3 mmHg which is within the sensors specifications.

2). P8, lines 170-176: The distance sensors were tested for accuracy using a carton disk mounted on a z-axis slide with a Vernier scale of 0.1mm per division (ZWG60, Misumi, Tokyo, Japan). The disk was elevated inside each cup from 0 mm (the rim of the cup) to a height of 37 mm within the cup with a maximum step of 5 mm. Each measurement was performed 30 times and for all the six cups. The average and standard deviation of the measurements were calculated. The maximum standard deviation for the measurements was 1.6 mm and the maximum absolute error of the mean values was 2 mm.

2. Regarding the outcome of the designed (and quantitatively monitored) therapy, further clarification can improve the manuscript. The authors claimed that the soft tissue compliance of the experimental group approached the values of the normal group after cupping for 4 weeks (lines 249-251). Does it mean that the difference in compliance values between the experimental group and the control group is not statistically significant after the 4 weeks? Thorough comparisons between the two groups after 4 weeks therapy will be helpful for the readers to understand the effect of the designed therapy.

Response: The author wish to thank the reviewer for his/her careful and rigorous review. No, the 4-week treatment did not significantly improve the soft tissue compliance of the myofascial group, only a trend toward the normal group was observed in Figs 4 and 5. In order to make readers understand the trend effect of treatment more clearly, we had added some descriptions in lines 271-275 of the revised manuscript.

Minor comments:

1. I recommend that the authors should change the title so that it can reflect their main contribution. For example, the word “baseball player” does not seem important enough to be emphasized in the title.

Response: We greatly appreciate the reviewer’s comment; the title has been revised to "Influence of quantified dry cupping on soft tissue compliance in athletes with myofascial pain syndrome."

2. Some sentences read awkward and seem to employ incorrect prepositions. (For example, the sentence starting from line 174 could be written much more concisely.) I recommend that the authors receive professional editing service.

Response: The manuscript has been read and corrected by an expert in English writing.

3. Only after I read the manuscript for a while, I could understand that the control group is also from the same baseball team. This fact should be clarified much earlier.

Response: Thank you for pointing this out. We added some wording instructions in line 25 of the Abstract section, and lines 86-87 and 96 of the Materials and methods section to clearly state that the normal group and the myofascial group were from the same baseball team.

4. The sentence in line 217-218 is very confusing. The current sentence reads that the tissue elevation reduced the pressure, which cannot be valid. The tissue elevation reduces the “magnitude of negative pressure”, or increases the absolute pressure value in the cup. The authors need to rephrase the sentence to clarify the meaning.

Response: Thank you for bringing this issue to our attention. In this sentence, we want to express the time effect of the pressure in the cup and the soft tissue being lifted during the 15-minute cupping period. We rewritten this sentence according to your suggestion to clearly express the meaning and avoid confusion for readers. The new sentences are as described in lines 233-235 of the revised manuscript.

5. The sentence in lines 220-222 implies that compliance should increase or decrease due to the mechanical input to the tissue. This cannot be valid. Compliance (or the inverse of the stiffness) is the mechanical property of any system like human tissue. Mechanical input and output (e.g., force and kinematics or vice versa) can be measured to estimate this property. Although the compliance might be nonlinear and change due to the input, the specific tendency cannot be assumed before the actual estimation. If the authors defined compliance correctly, the phrase following “because” in the mentioned sentence cannot be the cause of the significant increase of the compliance.

Response: Thank you for the reviewer’s reminder. To avoid misunderstandings caused by the description of this sentence, we rewritten this sentence as shown in lines 236-240 of the revised manuscript. In addition, we defined the concept of soft tissue compliance from the 15th References, which is provided for your reference.

6. The last statement starting from line 343 is clearly an overstatement. Either the suggested quantifying system or other more advanced system and the resulting quantified “dose” of cupping cannot “guarantee” the efficacy of cupping.

Response: The word "guarantee" is indeed inappropriate, thanks for the reminder. We have modified this sentence as described in lines 397-399 of the revised manuscript.

---

## [Decision Letter · Decision Letter 1]

2 Nov 2020

Influence of quantified dry cupping on soft tissue compliance in athletes with myofascial pain syndrome

PONE-D-20-15383R1

Dear Dr. Chen,

We’re pleased to inform you that your manuscript has been judged scientifically suitable for publication and will be formally accepted for publication once it meets all outstanding technical requirements.

Kind regards,

Yi-Hung Liao, Ph.D.

Academic Editor

PLOS ONE

Additional Editor Comments (optional):

Congratulations, I would like to appreciate the authors’ efforts to carefully revise the manuscript according to all reviewers' comments, and the quality of this work had been significantly improved. After careful evaluation, this manuscript should meet the criteria of publication.

Reviewers' comments:

Reviewer's Responses to Questions

**Comments to the Author**

1. If the authors have adequately addressed your comments raised in a previous round of review and you feel that this manuscript is now acceptable for publication, you may indicate that here to bypass the “Comments to the Author” section, enter your conflict of interest statement in the “Confidential to Editor” section, and submit your "Accept" recommendation.

Reviewer #1: All comments have been addressed

Reviewer #3: All comments have been addressed

2. Is the manuscript technically sound, and do the data support the conclusions?

Reviewer #1: Yes

Reviewer #3: Yes

3. Has the statistical analysis been performed appropriately and rigorously? 

Reviewer #1: I Don't Know

Reviewer #3: N/A

4. Have the authors made all data underlying the findings in their manuscript fully available?

Reviewer #1: Yes

Reviewer #3: Yes

5. Is the manuscript presented in an intelligible fashion and written in standard English?

Reviewer #1: Yes

Reviewer #3: Yes

6. Review Comments to the Author

Reviewer #1: The authors have modified the paragraphs accordingly and provided enough info to the questions. In addition, the authors addressed their ideas clearly and included adequate references. I really enjoy reading the article this time. I have no further suggestion or question to give. Many thanks for having me involved in the process.

Reviewer #3: (No Response)

7. PLOS authors have the option to publish the peer review history of their article (what does this mean?). If published, this will include your full peer review and any attached files.

Reviewer #1: **Yes: **Sheng-Ju Chuang

Reviewer #3: No

---

## [Editor Report · Acceptance letter]

5 Nov 2020

PONE-D-20-15383R1 

Influence of quantified dry cupping on soft tissue compliance in athletes with myofascial pain syndrome 

Dear Dr. Chen:

I'm pleased to inform you that your manuscript has been deemed suitable for publication in PLOS ONE. Congratulations! Your manuscript is now with our production department. 

Kind regards, 

on behalf of

Dr. Yi-Hung Liao 

Academic Editor

PLOS ONE